# Sensitivity Analysis and Optimal Design of a Stator Coreless Axial Flux Permanent Magnet Synchronous Generator

**Wenqiang Wang, Shaoqi Zhou \*, Hongju Mi, Yadong Wen, Hua Liu, Guoping Zhang and Jianyong Guo**

Department of Petroleum, Army Logistics University of PLA, Chongqing 401331, China; wqwang3035@gmail.com (W.W.); mimihj666@gmail.com (H.M.); elecrivalry@gmail.com (Y.W.); elecrivalry@gmail.com (H.L.); zgp064@gmial.com (G.Z.); gjy16889@gmail.com (J.G.)
\* Correspondence: shaoqizhou6308@gmail.com; Tel.: +86-023-8673-6189

**Abstract:** In this paper, the modified initial design procedure and economic optimization design of a stator coreless axial flux permanent magnet synchronous generator (AFPMSG) are presented to improve the design accuracy, efficiency, and economy. Static magnetic field finite-element analysis (FEA) is applied to the magnetic equivalent circuit (MEC) method to increase the accuracy of electromagnetic parameters and reduce the iteration times. The accuracy and efficiency of the initial design is improved by the combination of MEC method and static magnetic field FEA in the design procedure. For the economic optimization, the permanent magnetic (PM) material volume model, which affects the cost of the AFPMSG the most, is derived, and the influence degree of the main structure parameters, to the performance, is distinguished and sorted by sensitivity analysis. The hybrid genetic algorithm that combines the simulated annealing and father-offspring selection method is studied and adopted to search for the best optimization solution from the different influence degree and nonlinear interaction parameters. A 1 kW AFPMSG is designed and optimized via the proposed design procedure and optimization design. Finally, 3D finite-element models of the generator are simulated and compared to confirm the validity of the proposed improved design and the generator performance.

**Keywords:** stator coreless; axial flux permanent magnet synchronous generator (AFPMSG); hybrid genetic algorithm (GA); sensitivity analysis; economic optimization

---

## 1. Introduction

In the topic of permanent magnet electrical machine construction, recent works have shown that the usage of neodymium-iron-boron (NdFeB) and types of permanent magnet motor have drastically increased over the last decades, mainly due to cheaper cost and the demand of non-pollution energy. Although there are a lot of categories of permanent magnet generators available, the axial flux permanent magnet synchronous generator (AFPMSG), with a different electromagnetic path from traditional motors, is studied here. The AFPMSG has the merits of both a permanent magnet motor and a disc type motor, which have a small size, simple and compact structure, high power density and operation efficiency, and easy processing and manufacturing. Great application prospect and development value have been exploited in fields of electric cars, large industrial equipment, wind power generation, etc. [1–4].

The AFPMSG can be single-sided or double-sided, core or coreless, surface mounted or interior permanent magnet (PM), and have a single- or multi-staged configuration [5]. Among those types

of AFPMSG, double-sided AFPMSG with stator coreless has been widely researched and used in the industry. Compared with a stator core generator, the AFPMSG, without stator core, have the advantages of light weight, no cogging torque, high efficiency, and simple construction. The double-sided structure of the APFMSG has higher torque-to-volume ratio than that of other structures [6], and the greater the number of pole pairs that are designed, the higher ratio it will be. Besides, the negligible axial attraction force between the stator and rotor improves the reliable design of large diameter generators. However, because of the non-ferromagnetic stator core, the effective air gap length is larger and requires more active PM material to keep the magnetic field in the air gap, which increases the cost of the AFPMSG. Economic optimization design for the stator coreless AFPMSG becomes an important research topic.

With the further research and extensive application of AFPMSGs, the analysis of their design and optimization has been a hotpot research field recently. The non-uniform distribution of magnetic field, large air gap, and the interaction of structural parameters make the stator coreless AFPMSG a multi-variable non-linear system, which brings great difficulties to its optimization design [7]. At present, the design and optimization of the AFPMSG have been extensively studied and explored. The basic theory of AFPMSG, including the design procedure, performance calculation, and application, is presented in [8,9]. The design methods and development of performance analysis tools for axial flux permanent magnet machines is presented in [10]. Kahourzade, et al. and Huang, et al. [11,12] investigated the design and analysis of the AFPMSG with the sizing equation. In [13,14], analytical methods are used for the analysis and calculation of magnetic field in the generator. Study on the flux distribution of each part of the AFPMSG via magnetic equivalent circuit (MEC) is presented in [15,16]. A new three-dimensional finite element method based on improved Maxwell equations is proposed to calculate the no-load flux of an axial permanent magnet motor in [17]. To improve the design efficiency and accuracy, hybrid methods are used in the design, analysis, and calculation of the AFPMSG in [18,19]. However, the special structure and oversimplification would lower the design accuracy for the analytical method, and the 3D finite element method takes time to do the calculation and is inconvenient for design and analysis with multi-parameters varying in a large range. The calculation time and accuracy of the MEC method are moderate, and between the analytical method and finite element method. Additionally, as computer-aid technologies blossom, new or improved design methods are still under research to make the design more realistic.

Due to complex electromagnetic processes and practical application requirements, an optimal procedure is necessary to obtain better design results or to improve the performance of the motor for special needs. Recently, heuristic algorithms, especially the genetic algorithm (GA), have been used in motor optimization [20,21]. The development of field computation and multi-objective optimal design in electromagnetics were studied in [22,23]. Performance optimizations, including the maximum power density, low cogging torque, minimum material cost, etc., via the GA, are presented in [24–26]. However, in these studies, the sensitivity analysis of the generator design parameters to output performance and economy optimization is not paid enough attention, which affects the efficiency and results of the motor optimization design.

Considering the special requirements for the design and application, improved design procedure and economic optimization of a stator coreless AFPMSG are proposed in the paper. The field-circuit method, combining the MEC method and static magnetic field FEA, is used in the initial design procedure to improve the design efficiency and accuracy. An active PM material volume model, which concerns the cost of the AFPMSG the most, is derived for the economical optimal design. In order to find out the influence degree of each parameter, sensitivity analysis on the main design parameters to the performance and material volume consumption of the AFPMSG is carried out. Considering the main design parameters' non-linear interaction and different influence degree to the output performance, father–offspring selection and a simulated annealing method are applied in the GA to improve the design accuracy, globality, and fast convergence. Improved GA is used to perform the calculation of the AFPMSG economic optimization model for the best solution to reduce active

material volume cost and increase the economy on the condition of keeping the electrical performance unchanged. The schematic overview of this paper is showed in Figure 1.

In the following, basic structure and equations are introduced and the improved initial design procedure of the AFPMSG with the field-circuit method is given in Section 2. The economic optimization model, that is the active PM material volume, is derived and studied in Section 3, as well as the sensitivity analysis about the main design parameters to the performance and material volume consumption of the AFPMSG. Section 4 introduces the improved GA for the optimal design of the APFMSG, which combines the simulated annealing and father-offspring selection method. In Section 5, the validity of the improved initial design procedure and economic optimization design are evaluated by comparison and 3D FEA. Finally, conclusions are drawn in Section 6.

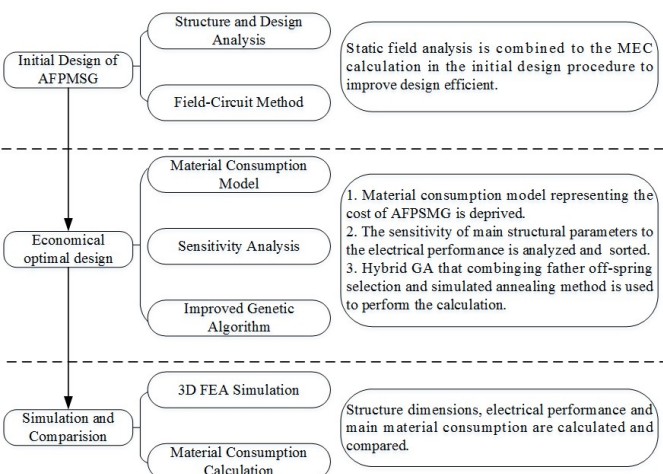

**Figure 1.** The schematic overview of this paper.

## 2. AFPMSG and Initial Design

### 2.1. Structure and Design Analysis

Considering the advantages of the axial flux permanent magnetic motor and the stator coreless structure, a dual-rotor AFPMSG with coreless stator was studied in this paper. The considered AFPMSG consists of two outer rotors and a coreless stator clamped in the middle. Sector permanent magnets are placed on the rotor yoke surfaces which formulates the rotor structure. The stator is made of coils and non-ferromagnetic materials, and the non-overlapping concentrated coils are held together and supported by non-ferromagnetic materials. The structure of AFPMSG is shown in Figure 2.

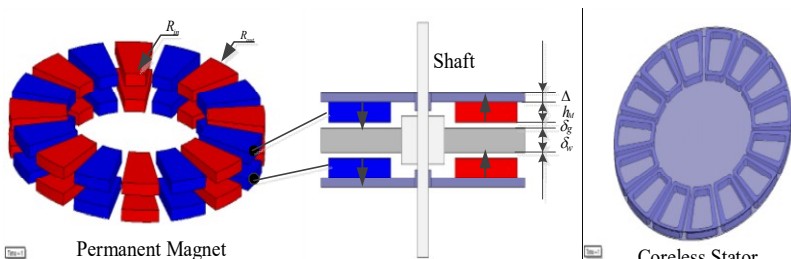

**Figure 2.** Illustration of main structure and dimensions of axial flux permanent magnet synchronous generator (AFPMSG).

Electromagnetic relation and sizing equations are the basics of motor design. The design of the AFPMSG follows the fundamentals is presented in [5]. The main sizing equation of the AFPMSG that shows the relationship between the output power and generator structure parameters is given by:

$$P_e = \frac{\pi^3}{32} n_s k_w A_{av} \alpha_i B_\delta (D_{out} + D_{in}) \left( D_{out}^2 - D_{in}^2 \right) \tag{1}$$

where, $k_w$, $n_s$, $A_{av}$, $B_\delta$, $D_{out}$, and $D_{in}$ are winding factor, rotor speed, average electrical loading, flux density of the air gap, and outer and inner diameter of the AFPMSG, respectively; and $\alpha_i$ is the calculated polar arc coefficient. For the sector-shaped permanent magnet structure, the calculated polar arc coefficient is equal to the polar arc coefficient $\alpha_p$ which is the ratio of magnet width to pole pitch.

When designing the AFPMSG, appropriate design values should be selected for some design quantities according to experience and design requirements, such as electromagnetic load, inner-to-outer diameter ratio, etc., and the final value should be determined after checking. Electrical load represents the number of ampere conductors per unit circumference along the armature surface, and magnetic load, namely air gap flux density, refers to the average flux density along the air gap surface when no load occurs. Electromagnetic load equations are expressed in:

$$A(r) = \sqrt{2} m N I_a / \pi r \tag{2}$$

$$B_\delta = \varnothing_m / \alpha_p \tau L_{ef} \tag{3}$$

where, $m$, $N$, and $I_a$ are the number of the phases, coil turns per phase, and rated current of armature winding, respectively. $\varnothing_m$, $\alpha_p$, $\tau$, and $L_{ef}$ are main flux per pole, polar arc coefficient, magnetic pole pitch, and effective length of armature winding, respectively. It is obvious that the electric load is a function of radius, and the maximum electric load is at the inner diameter for the AFPMSG. In order to avoid local overheating, it should be designed according to the maximum electric load. Besides, the air gap magnetic field is the medium of energy conversion of a generator; therefore, the design of magnetic load has a significant impact on the performance of a generator. Because of the complicated flux distribution, the value is given by estimation in the initial design [27].

The diameter ratio $\gamma$ is used to present the relation between inner and outer diameter, and the relation is given by:

$$\gamma = D_{in} / D_{out} \tag{4}$$

The diameter ratio $\gamma$ is an important design parameter affecting the performance and economy of the AFPMSG. Based on the extreme value analysis and comprehensive consideration of copper loss, cost, etc., the value of $\gamma$ is usually 0.45–0.60 [5]. However, the actual value depends on the design objectives and optimization.

The energy conversion of the motor is carried out in the air gap, hence the length of the air gap affects the structure and performance of a generator. Due to the non-ferromagnetic stator structure, the air gap length of the AFPMSG is made up of the double-sided air gap and the stator, as expressed in Equation (5). Additionally, the axial length of the AFPMSG can be obtained by Equation (6).

$$L_{air} = 2\delta + \delta_w \tag{5}$$

$$L = L_{air} + 2h_M + 2h_r \tag{6}$$

where $\delta$, $\delta_w$, $h_M$, and $h_r$ are the air gap length, thickness of stator, PM, and rotor, respectively. $L_{air}$ and L are the effective air gap length and axial length of the APFMSG, respectively.

Studies on the performance of the APMFSG with different types of coil winding are presented in [28] and specifications can be obtained by analytical calculation. Considering the non-ferromagnetic stator structure that increases the effective length of the air gap, the thickness of winding should be decreased as much as possible. On the contrary, too small of a space for the stator windings will lead to

heat dissipation problems. In the AFPMSG, the inner radius is the most limited space to place the stator windings and thus the space factor is introduced from the stator core motor to calculate the thickness of the stator winding appropriately. The thickness of the armature winding coil is determined by the occupancy rate of the total cross-sectional area of armature winding coil at the inner diameter. As shown in Equation (7), the initial thickness of stator can be calculated when a proper space factor is selected.

$$S_f = 4mNaa'D^2/(\pi D_{in}\delta_w) \tag{7}$$

where $N$, $a$, $a'$, and $D$ are the number of turns in series for each phase winding, the number of parallel branches, coils per turn wound around the conductor, and the diameter of the wire, respectively.

Another important step is to calculate the loss and efficiency of the AFPMSG accurately in the design procedure. Due to the absence of a stator core, the loss of the AFPMSG consists of copper loss, $P_{cu}$, eddy current loss, $P_{eddy\_cu}$, mechanical loss, $P_{mech}$, and rotor core loss, $P_{core}$. The calculation of every aspect loss of the AFPMSG has been researched and presented in [5,29,30]. Thus, the efficiency of the AFPMSG is expressed as:

$$\eta = P_{out}/\left(P_{out} + P_{cu} + P_{eddy\_cu} + P_{mech} + P_{core}\right) \tag{8}$$

### 2.2. Improved Initial Design Procedure

In the design procedure of the AFPMSG, as discussed in the previous part, some design parameters were determined by experience and estimating. Considering the characteristics of the structure and magnetic field distribution, the analytical design and FEA may be either inaccurate for special magnetic field distribution or time-consuming, due to too many parameters, for simulation. In this paper, the MEC method, combining the static magnetic field FEA, was adopted to improve the design efficiency and accuracy. The way it was used in the paper makes full use of the advantages of the static magnetic field FEA (accuracy and less time-consuming) to both solve the electromagnetic parameters' inaccuracy problem and reduce the number of iterations in the MEC method. The MEC method in this paper converts the non-uniform spatial magnetic field into the equivalent multi-section magnetic circuit, and approximates that the magnetic flux in each section is uniformly distributed along the cross-section and length, and transforms the calculation of magnetic field into the calculation of magnetic circuit [8]. Figure 3 shows the magnetic circuit and calculation is given by:

$$H_1L_1 + H_2L_2 + \cdots + H_nL_n = F \tag{9}$$

where $H_i$ and $L_i$ are the magnetic field intensity and path length of the $n$-th magnetic circuit, respectively.

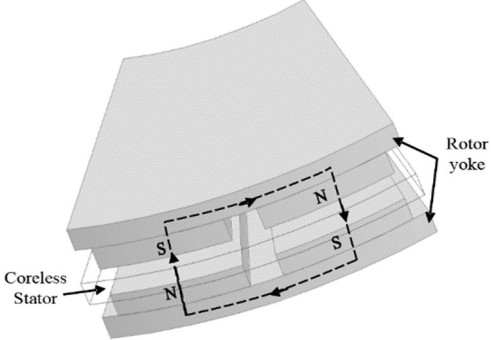

**Figure 3.** Flux paths in 3D plane for the stator coreless AFPMSG.

In order to increase the accuracy of design parameters, finite element analysis was adopted in the design procedure, and the static magnetic field analysis of the AFPMSG was mainly used to obtain

the electromagnetic parameters that were difficult and inaccurate for circuit calculation. Professional electromagnetic field simulation software, Ansys Maxwell, was used to establish the model and conduct the finite element analysis. Figure 4 shows the 2D view of the magnetic field distribution of the AFPMSG in the mean radius. The magnetic field distribution and leakage flux can be obtained through the static field analysis.

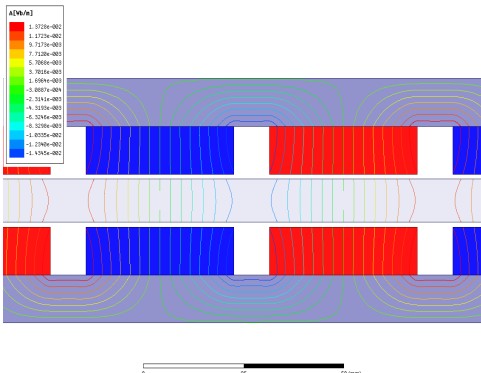

**Figure 4.** 2D view of the magnetic field distribution of AFPMSG.

Based on the design analysis, the improved design procedure with the field-circuit method was proposed, as shown in Figure 5, and the stator coreless AFPMSG rated at 1 kW, 230 V (phase), three-phase, 50 Hz, and 300 r/min was designed. The classical structure mode was adopted in the design of the rotor pole structure and stator winding. Design parameters were chosen properly with comprehensive consideration. Table 1 lists the detailed design results of the AFPMSG.

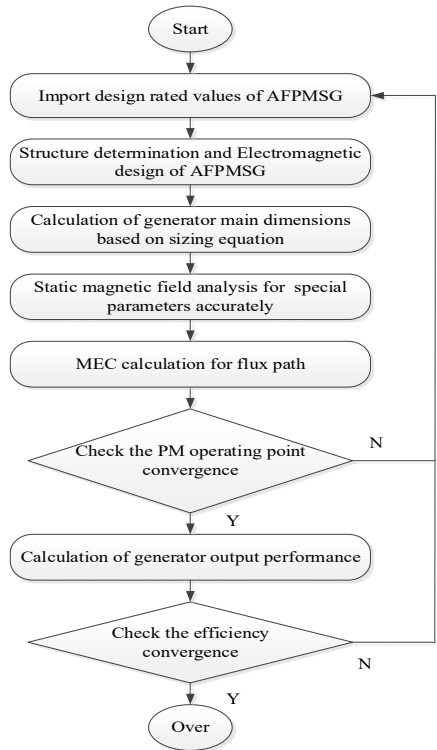

**Figure 5.** Improved design procedure of the stator coreless AFPMSG.

**Table 1.** Initial design results of the AFPMSG.

| Design Parameters | Values |
| --- | --- |
| Outer diameter (mm) | 380 |
| Inner diameter (mm) | 210 |
| Thickness of permanent magnet (PM) (mm) | 10 |
| Pole arc coefficient | 0.8 |
| Air gap length (mm) | 1.2 |
| Thickness of stator (mm) | 9 |
| Thickness of rotor yoke (mm) | 10 |

*2.3. Performance Analysis*

3D finite element analysis was adopted to simulate the output performance of the AFPMSG through Ansys Maxwell 16.0 software. Ansys Maxwell is the most practical electromagnetic analysis software at present, which can analyze and calculate various electromagnetic fields and multi-state system problems with its advanced adaptive meshing technology and convenient self-defined material library [31]. The main steps of the simulation analysis are shown in Figure 6 [32]. In this paper, the establishment of the model was based on the design value of the generator structure dimension. The other settings and conditions were set as the normal or rated value.

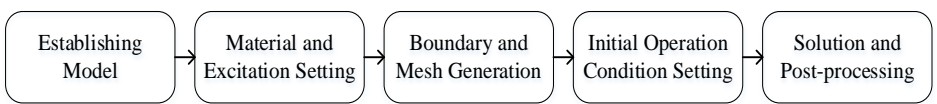

**Figure 6.** The simulation and design flow chart of Ansys Maxwell.

The performance analysis of the generator mainly focused on the analysis and research of the induced electromotive voltage, torque, and efficiency under load to verify the design results. The initial simulation model of the AFPMSG was established based on the initial design structure dimension values, as shown in Figure 7a. The initial operation condition was under a rated constant speed of 300 r/min with a three-phase symmetrical-rated resistive load. The output performance curve, including the induced voltage and torque of the AFPMSG, could be obtained through simulation, as shown in Figure 8.

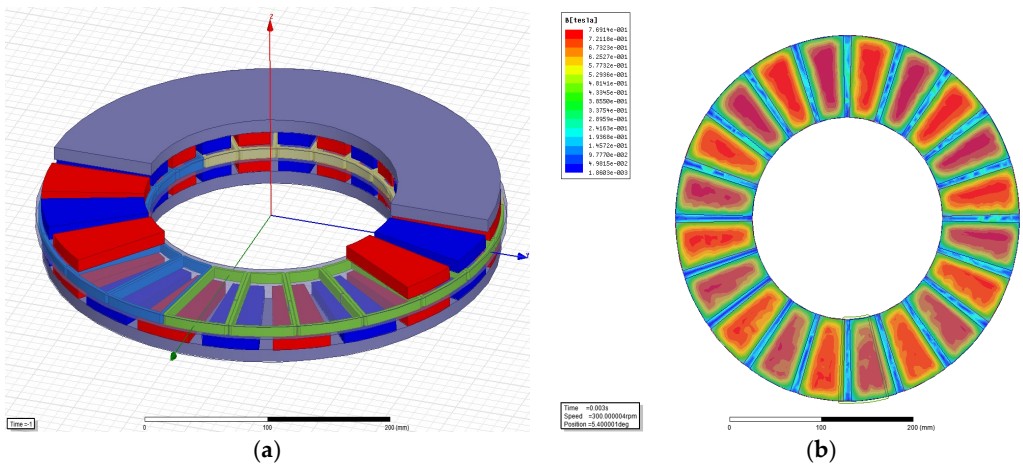

(**a**)                                  (**b**)

**Figure 7.** 3D model and air gap magnetic density of the AFPMSG. (**a**) 3D model of the AFPMSG; (**b**) Magnetic flux density (Mag B) distribution in the air gap.

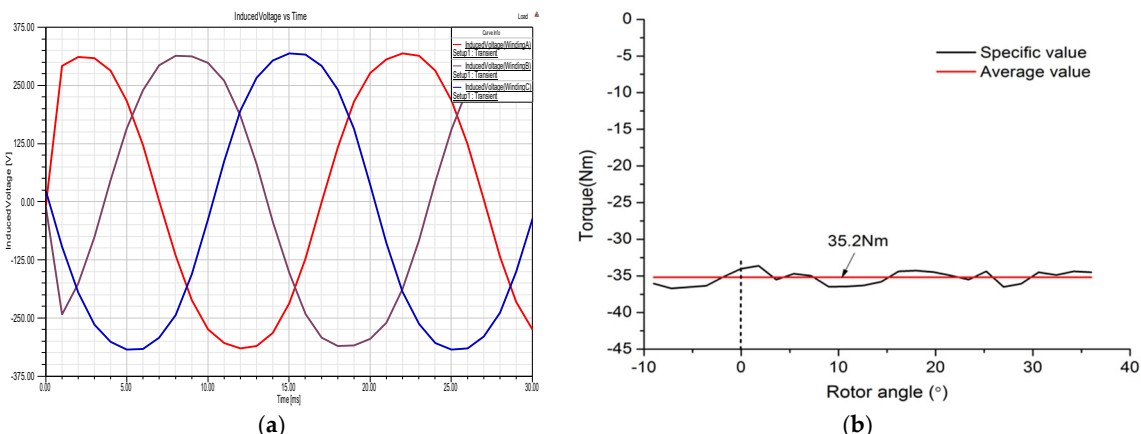

**Figure 8.** Output performance of the AFPMSG. (**a**) Output-induced voltage waveform; (**b**) torque against the rotor angle.

In Figure 7b, the flux density on the middle plane of the effective air gap is illustrated. The air gap magnetic density decreased from the center of the magnetic pole to both sides. The phase-induced voltage and torque curves are presented in Figure 8a,b. It was evident that the AFPMSG had a sinusoidal-induced voltage and relatively stable electromagnetic torque output. The output-induced voltage was stable with low harmonics. The electromagnetic torque ripple caused by eddy current and error was small, and overall was balanced in a cycle, which can be neglected. The maximum amplitude of the torque fluctuation was about 1.5 Nm. At rated speed, the torque was about 35.2 N.m, the output power was 1.105 kW, the phase voltage was about 322.54 V, and the efficiency was 90.3%. The results showed good agreement with the basic design requirements. However, as parts of the design parameters were designed with empirical value, the design results could still be further optimized for actual practical needs.

## 3. Optimization Model and Sensitivity Analysis

### 3.1. Optimization Model of the AFPMSG

The optimization of a motor includes the performance, economy, and other special needs. Structure parameters are the main optimization objects of a motor, and optimization is to adjust the structural parameters to get the best results so that some performance or indicators of the motor can be optimized. Under the condition of keeping the shape and structure of the AFPMSG unchanged, the main structural parameters affecting the performance and economy were as follows: inner-to-outer diameter ratio, $\gamma$, effective air gap length, $\delta$, the thickness of stator, $\delta_w$, the thickness of PM, $h_M$, rotor yoke thickness, $h_r$, and polar arc coefficient, $\alpha_p$. When $\gamma$ increased, the effective material consumption and performance of the AFPMSG decreased, while the economy improved. In order to maintain the flux density of the air gap, the longer the effective air gap was, the more PM materials were needed, which would make the generator larger. Additionally, the thickness of the permanent magnet and rotor yoke was mainly to meet the needs of the air gap magnetic field of the AFPMSG, to ensure that the magnetic circuit was set at the saturation critical point and the yoke structure was not deformed.

The first priority of optimization is determining the optimization objective. According to Section 2, the electrical performance of the initially designed AFPMSG could basically meet the design requirements, but the design of the structural parameters were rough and the economy of the AFPMSG could be improved. The total material cost of the AFPMSG could obtained from the structure and material used in the generator, as expressed by:

$$C_{m,total} = C_{cu}m_{cu} + C_{Fe}m_{Fe} + C_{PM}m_{PM} \tag{10}$$

where $m_{cu}$, $m_{Fe}$, $m_{PM}$, $C_{cu}$, $C_{Fe}$, and $C_{PM}$ are, respectively, the mass and price of the copper, steel, and PM material. The mass of each material was computed based on the material volume and the corresponding mass density. In the calculations, the material price of copper, steel, and PM were considered equal to 15, 3, and 80 Ero/kg, respectively [29,33].

In view of the material price, the PM material cost was the main part of the production cost for the stator coreless AFPMSG. Meanwhile, the processing and manufacturing of PM was much more difficult, and it was easy to waste PM material, which lowered the economy of the AFPMSG. Therefore, the objective function of PM volume consumption was established for economic optimization in this paper, as given below.

$$min : f(X) = \frac{\pi}{4}\alpha_p D_{out}^2\left(1 - \gamma^2\right)h_M \tag{11}$$

The optimization of the AFPMSG was carried out on the premise of ensuring that the electrical performance and external structure remained unchanged. Therefore, the outer diameter and axial length was taken as the structure constraints, while the electrical performance parameters, such as flux density of air gap and rotor yoke, efficiency, etc., were taken as the electrical requirements. The constraints function could be obtained as follows:

$$\begin{cases} g_1(X) = \eta_0 - \eta \leq 0 \\ g_2(X) = B_{\delta 0} - B_\delta \leq 0 \\ g_3(X) = B_{r0} - B_{ro0} \leq 0 \\ g_4(X) = D_{max} - D_0 \leq 0 \\ g_5(X) = h_{max} - h_0 \leq 0 \\ g_6(X) = P_0 - P \leq 0 \end{cases} \tag{12}$$

where $\eta_0$, $B_{\delta 0}$, $B_{r0}$, $D_0$, $h_0$, and $P_0$ are the initial design value of efficiency, flux density of air gap and rotor yoke, size of generator diameter, PM thickness, and output power, respectively.

Therefore, the multi-parameter AFPMSG economic optimization problem can be transformed into a mathematical model, with the volume consumption of permanent magnets as the objective function.

$$\begin{cases} min : f(X) = \frac{\pi}{4}\alpha_p D_{out}^2\left(1 - \gamma^2\right)h_M \\ X = \left(D_{out}, h_r, \alpha_p, \gamma, h_M, \delta \right)^T \\ s.t. B_\delta > B_{\delta 0}; h_{max} \leq h_0; D_{max} \leq D_0 \\ \eta \geq \eta_0; B_{r0} \leq B_{ro0}; P \geq P_0 \end{cases} \tag{13}$$

*3.2. Sensitive Analysis for Design Parameters*

The optimization design of the AFPMSG was a process of repeatedly modifying parameters. In order to eliminate the factors that had little influence on the objective function and reduce the time spent in modeling and optimization, it was necessary to examine the correlation between design parameters and objective functions by sensitivity analysis before optimization began.

The optimum design of the AFPMSG was performed based on ensuring the demand of electrical performance. Material consumption and economy of the AFPMSG are also affected by the relationship between electrical performance and structural parameters. Therefore, the sensitivity of the AFPMSG structure parameters to the performance were mainly analyzed in this paper. Considering the complexity and inaccuracy of the AFPMSG design calculation, this paper combined analytic calculation with ANSYS simulation. Assume the performance function is $y = f(X)$, the AFPMSG parameter is $X$, their range of variation is $\left[x_i', x_i''\right]$, and the corresponding performance dependent variable is $y$, the range of variation $y$ is $\left[y_i', y_i''\right]$. The relative variation of the dependent variable and independent

variable are $\Delta X/X$ and $\Delta y/y$, respectively. The sensitivity of the AFPMSG performance to the parameters is expressed as follows:

$$S_X^Y = \frac{\partial y}{y} \Big/ \frac{\partial X}{X} = \frac{X}{y}\frac{\partial y}{\partial X} \tag{14}$$

For linear $y = f(X)$, the partial derivative $\partial f/\partial x_i$ is constant, and the range of performance independent variables varies:

$$y' = f(u') = a_0 + \sum_{i=1}^{n} a_i u_i' \tag{15}$$

$$y'' = f(u'') = a_0 + \sum_{i=1}^{n} a_i u_i'' \tag{16}$$

where in the formula $a_i$ is a linear coefficient and $u_i$ is an extreme point. The corresponding range of variation to $x_i$ is $[u_i', u_i'']$. The performance function of the AFPMSG structural parameters is a multi-variable non-linear function. In this paper, the differential sensitivity $s_{X_i}^y$ is obtained by taking linear approximation at the design value $X_0$, and the variation range of the dependent variables $y$ is obtained.

$$y' \approx y_0 + \sum_{i=1}^{n} S_{X_i}^y \left(u_i' - x_{i0}\right) \tag{17}$$

$$y'' \approx y_0 + \sum_{i=1}^{n} S_{X_i}^y \left(u_i'' - x_{i0}\right) \tag{18}$$

where $y_0 = f(X)$. The influence of different parameters on the AFPMSG performance can be obtained from the above formula.

In order to analyze the sensitivity of the structural parameters to the performance of the dual rotor coreless AFPMSG, analytical calculation and finite element simulation were used, due to the complicated flux distribution and the interaction of multi parameters. Figure 9 shows the sensitivity of the main structural parameters to the electrical performance, including inner-to-outer diameter ratio, $\gamma$, air gap length, $\delta$, thickness of PM, $h_M$ and rotor yoke, $h_r$, and the pole arc coefficient, $\alpha_p$.

It can be seen that some structural parameters, like diameter ratio, $\gamma$, and PM thickness, $h_M$, improved the output performance with the increasing of parameter value, while parameters such as air gap length, $\delta$, decreased the performance as the parameter value increased. As shown in Figure 9a,e, there was a critical vale of the diameter ratio, $\gamma$, and rotor thickness, $h_r$, before which the relation between them and the output performance of the AFPMSG was nearly linear. Form Figure 9b, increasing the air gap length reduced the rated output of the AFPMSG and the utilization ratio of the permanent magnet, but the appropriate air gap length was conducive to reduce the torque fluctuation and make the generator run more smoothly. The increase of pole arc coefficient, $\alpha_p$, added the area of PM and magnetic flux of each pole, but at the same time also increased the leakage flux between each magnetic pole. The influence of pole arc coefficient, $\alpha_p$, to the output performance of the AFPMSG is shown in Figure 9c. It is obvious that the relationship between PM thickness and generator performance was monotonically increasing in Figure 9d. Moreover, the influence degree of the AFPMSG structural parameters on the output performance and PM material consumption could be obtained and arranged in sequence as follows: air gap length, permanent magnet thickness, pole arc coefficient, internal and external diameter ratio, and rotor yoke thickness. The sensitivity analysis of the structural parameters obtained by simulation was consistent with the theoretical analysis, which is helpful to understand and optimize the AFPMSG.

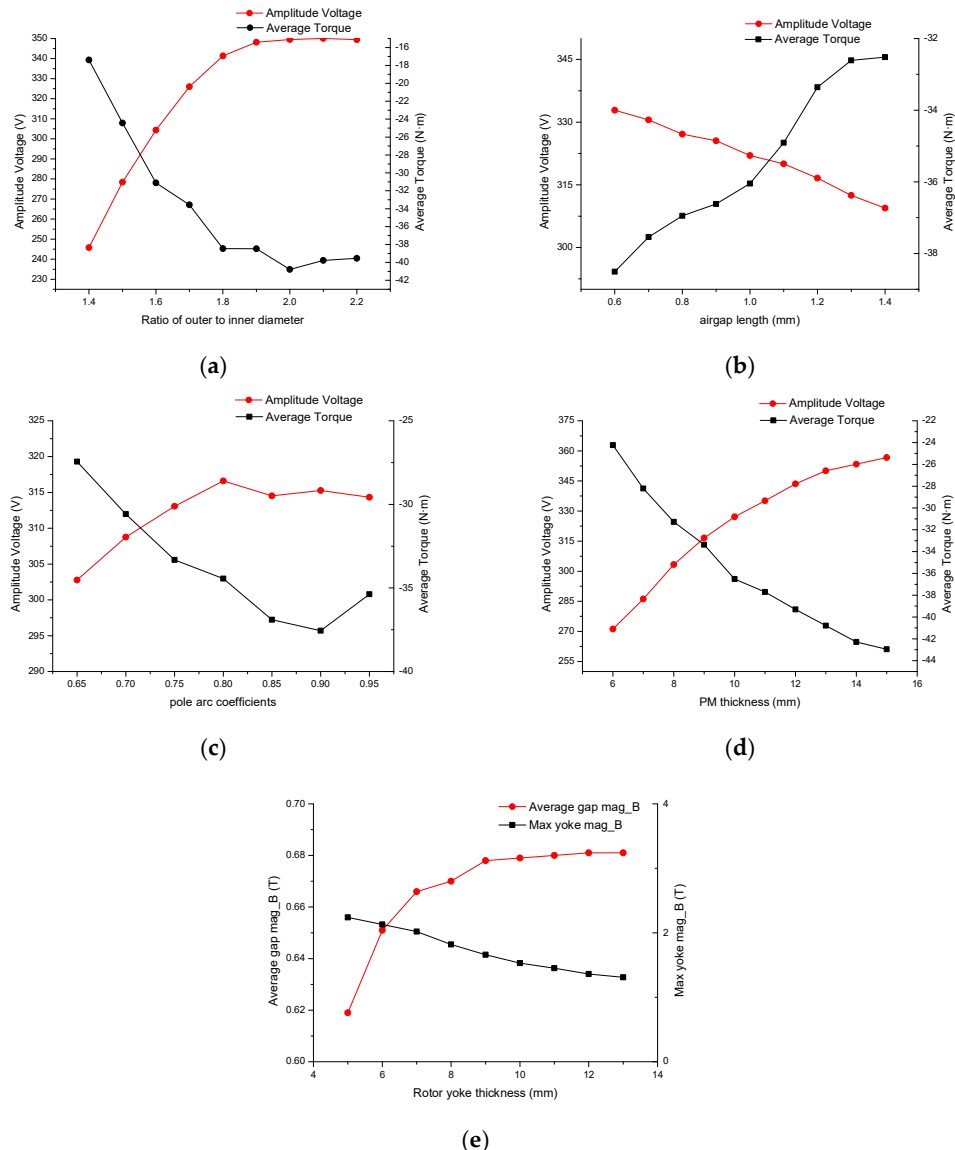

**Figure 9.** The sensitivity of the main structural parameters to the electrical performance. (**a**) The sensitivity of diameter ratio, $\gamma$, to (**b**) the sensitivity of air gap length, $\delta$, to the output performance of the AFPMSG, (**c**) the sensitivity of pole arc coefficient, $\alpha_p$, to (**d**) the sensitivity of PM thickness, $h_M$, to the output performance of the AFPMSG; (**e**) the sensitivity of rotor yoke, $h_r$, to the output performance of the AFPMSG.

## 4. Improved Genetic Algorithm

Multi-parameters and the interaction between parameters make the design and optimization of the AFPMSG more complicated. It is difficult to optimize the design and improve the economy on the basis of ensuring the output electrical performance. As discussed previously, the economic optimization of the AFPMSG is a multi-parameter and non-linear optimization process. The selection of the optimization algorithm has a significant impact on the optimization results and efficiency. Multi-parameter non-linear optimization algorithms are mainly the direct search method and random search method. Modern heuristic motor optimization methods, such as genetic algorithm (GA) and simulated annealing (SA), overcome the shortcomings of initial conditions, local optimization, and difficult multi-dimensional considerations, and have been rapidly developed and applied. In this paper, a hybrid genetic algorithm, which combines the simulated annealing method and father–offspring hybrid selection method, is proposed based on a traditional GA. GA is a random search algorithm

based on the natural selection and natural genetic mechanism to search the optimal solution of the problem [34,35]. It has the advantages of simple process, high degree of parallelism, strong global search ability, and scalability.

SA comes from the simulation of cooling in solid annealing. Its basic idea is to use a thermodynamic system to represent the optimization problem of demand solution, and to simulate the optimization process by gradually cooling the system to the lowest energy state. In the process of optimization, it has the ability of a probability jump, which can avoid falling into the local optimum solution, and makes up for the deficiency of GA in local optimization. In this paper, exponential annealing was used to reduce the temperature, as shown in Equation (19), and the Metropolis criterion shown in Equation (20) was used to judge whether to accept new solutions or not. The iteration process of "generating new solutions, judging, accepting or abandoning" was realized to find the optimal solution at this temperature.

$$T_i = T_0 k^{i-1} \tag{19}$$

$$p = \begin{cases} 1, \ f(Y') \geq f(Y) \\ exp\left\{ \frac{-[f(Y)-f(Y')]}{T_i} \right\}, \ f(Y') < f(Y) \end{cases} \tag{20}$$

where $T_i$ is the temperature of the current annealing process, $k$ is the attenuation coefficient, $T_0$ the initial stability, and $Y$ and $Y'$ are the current solution and the new solution, respectively. $p$ is used to determine whether the new solution is acceptable or not.

The father–offspring selection method improves the algorithm and adds stochastic factor control parameters so that the best individuals of the previous generation can be inherited, unconditionally, to the next generation. Therefore, the fitness sequence must be monotonic, that is, the evolutionary algebra is monotonic. The algorithm is the execution strategy of the genetic algorithm to ensure convergence, and it solves the problem that the traditional genetic algorithm may miss the global optimum, etc. The steps to implement the method were as follows:

Step 1: (Initialization) Random generation of the initial population of $M$ pairs, $X(0)$, $t = 0$;

Step 2: Population evolution:

(1) Generate intermediate population $Y(t)$ by crossing $M$ pairs in $X(t)$ independently;
(2) Execute mutation independently for each intermediate individual in $Y(t)$ to generate progeny population $Z(t)$;
(3) Select M pairs' mother population as the new generation population $X(t+1)$ from the union of parent population $X(t)$ and offspring population $Z(t)$;

Step 3: End and output the optimal individual in $X(t+1)$ after termination criterion is satisfied.

In this paper, improved GA was proposed for the problems that the algorithm itself has such as slow convergence and premature convergence. SA was applied in the fitness function to improve convergence speed and avoid premature convergence, while the father–offspring hybrid selection method was used to ensure that the best individuals of the previous generation could be inherited to the next generation, to realize elite legacy and improve global accuracy. The improved GA calculation flowchart is shown in Figure 10.

The main steps of the improved hybrid GA were as follows:

Step 1: Set up the initial parameters, such as generator variables, random control parameters, population size, $N$, maximum evolutionary algebra, $M_{max}$, initial and minimum annealing temperatures, $T_0$ and $T_{min}$, and temperature attenuation coefficient, $k$. Initial population is generated randomly.

Step 2: Carry out fitness calculation and evaluation, and annealing operation is adopted to produce new solutions. Using Metropolis criterion to judge whether to accept or not, and according to the maximum number of iterations and the minimum annealing temperature to select the calculation program, if not satisfied, go to step 3; when satisfied, stop iteration, jump to step 4.

Step 3: Renew the annealing temperature, and obtain the intermediate generation population through genetic, crossover, mutation, and other operations. Combine the parent population with the intermediate population; then select and arrange individuals according to fitness and form of the next generation population with individuals generated randomly. Go to step 2.

Step 4: Take the first individual in the current population as the optimal solution of the problem, and the calculation ends.

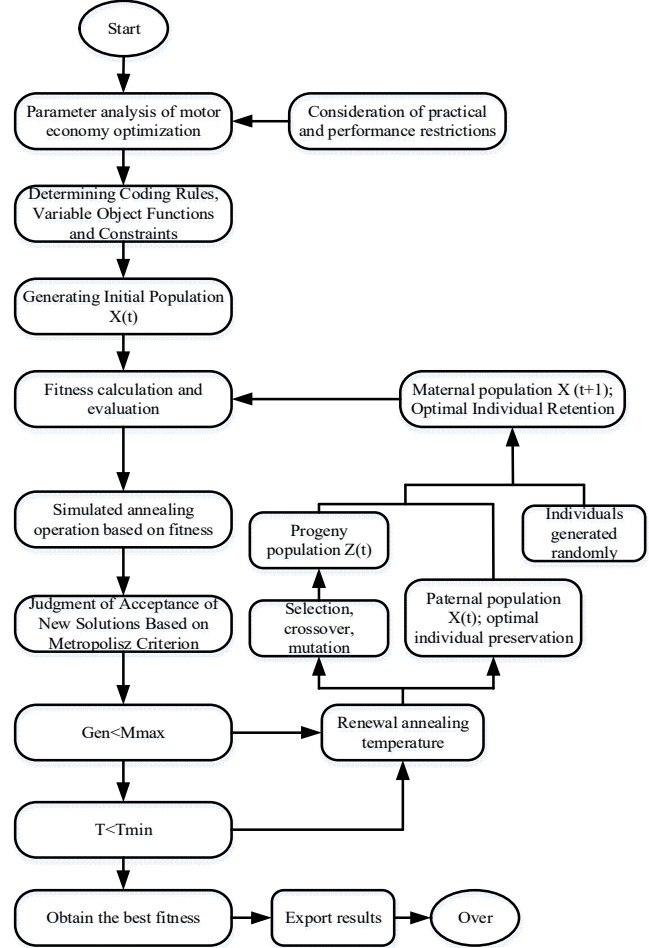

**Figure 10.** Flowchart of hybrid genetic algorithm (GA).

## 5. Optimal Design and Finite Element Analysis

Based on the initial design scheme and optimization model of the AFPMSG, the structural parameters of the generator were optimized by using the hybrid genetic algorithm programmed by Matlab. The optimized dimension parameters were obtained and compared as shown in Table 2.

**Table 2.** Improved dimensions of AFPMSG.

| Structural Parameters of AFPMSG | Values | |
|---|---|---|
| | **Initial Design** | **Optimized Design** |
| Outer diameter (mm) | 380 | 380 |
| Inner diameter (mm) | 210 | 227.5 |
| Thickness of PM (mm) | 10 | 9.3 |
| Pole arc coefficient | 0.8 | 0.77 |
| Air gap length (mm) | 1.2 | 1 |
| Thickness of stator (mm) | 9 | 9 |
| Rotor yoke thickness (mm) | 10 | 8.6 |

Finite element analysis (FEA) was proceeded on the AFPMSG to evaluate the improved design presented in the previous sections. 3D models of AFPMSG were developed and simulated by Ansys Maxwell 16.0 software based on the improved dimensions of the AFPMSG in Table 2, as shown in Figure 7. The models before and after optimization were set and simulated with the same conditions. The electrical performance was the first concern, and the consumption of the main material was the target for optimization. Figure 11 shows the distribution of the magnetic flux density along the circumference and comparison of the magnetic flux density (Mag B) curve under one pole arc. It can be seen that the distribution of Mag B presented a flat top wave under one pole arc and was periodically distributed along the circumference direction. The average value of Mag B changed little between the initial design and optimal design. As mentioned in Section 4, the decrease of the rotor yoke thickness increased the value of the max flux density in the rotor yoke, while the air gap flux density almost remains unchanged under the integrated influence of the air gap length and PM thickness decreasing.

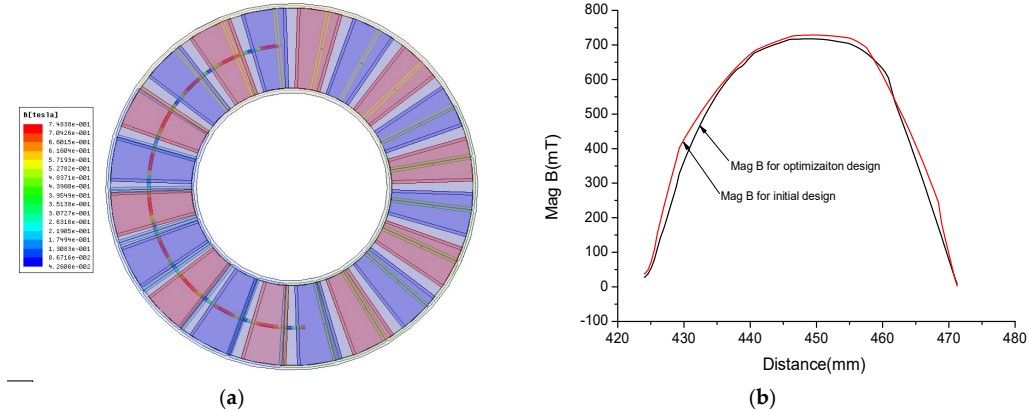

(**a**)                    (**b**)

**Figure 11.** The simulation results of Mag B. (**a**) The distribution of Mag B along the circumference direction; (**b**) the comparison of Mag B curve under one pole arc.

Three-phase voltage and torque of the generator are important aspects of output performance, which are discussed in the paper. The induced voltage and torque of the optimal design under rated load was achieved and compared with the initial design, as shown in Figure 12. Obviously, the output phase voltage of the optimal design still had a good sinusoidal curve and the torque performed smoothly. The comparison of the induced voltage and torque curves showed that the output performance of the AFPMSG after optimization was almost the same with the initial design, which can satisfy the requirements of the design purpose. Besides, the results were consistent with the comparison of Mag B.

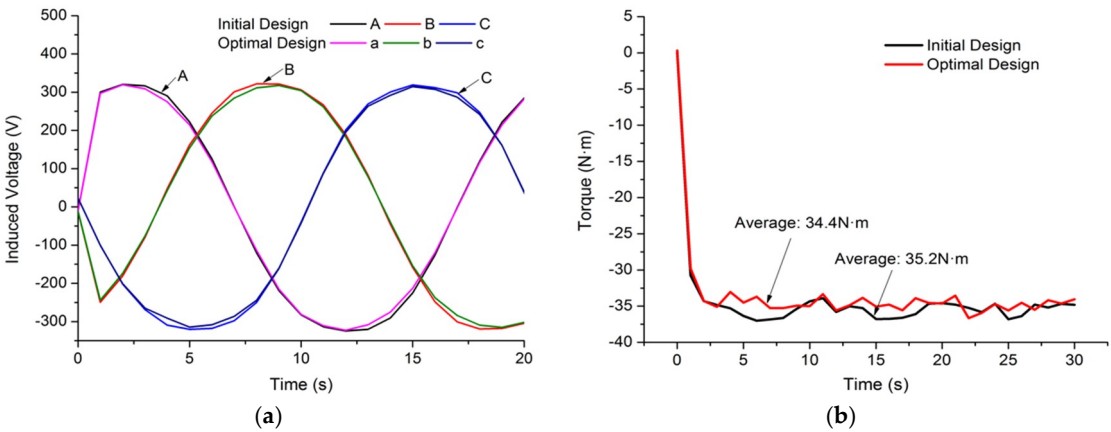

(**a**)                    (**b**)

**Figure 12.** The comparison of the AFPMSG output performance. (**a**) Output-induced voltage waveform; (**b**) output torque waveform.

The economy of AFPMSG was improved through the consumption optimization of effective material, especially the PM, and the comparison of the electrical performance and the main material consumption are shown in Figure 13. The efficiency and output power of the AFPMSG also changed little, while the volume of the main material reduced a lot, as the calculation results shown in Table 3. It was observed that the whole volume of the rotor yoke and PM for the AFPMSG was reduced obviously after optimization, 19% and 15.8%, respectively. The results show that economic optimization of the AFPMSG, through the main material volume cost model and improved GA method, can be achieved on the condition of keeping the electrical performance unchanged. It was clear that the 3D FEA results confirmed the economic optimization of the AFPMSG. The validity of both the improved initial design procedure with MEC, combining the static magnetic field FEA, and the economic optimization design through main material consumption model and improved GA, were confirmed by the results and analysis in the paper, which can be applied or used as a reference for better design. In addition, the research of this paper helps a lot for the further understanding of the AFPMSG.

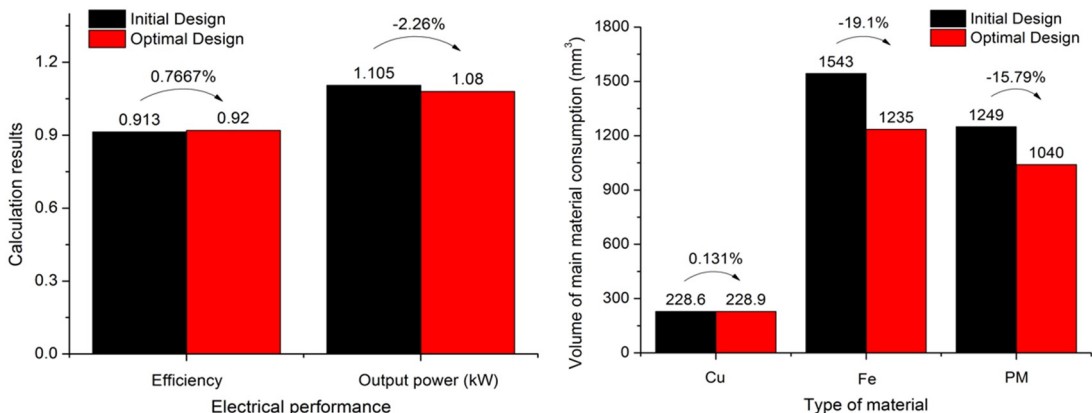

**Figure 13.** The comparison of electrical performance and main material consumption.

**Table 3.** Comparison of output performance and main material consumption.

| Generator Parameters | Before Optimization | After Optimization | Change Rate |
|---|---|---|---|
| Induced voltage (V) | 322.54 | 316.06 | −2% |
| Average torque (N·m) | 35.2 | 34.4 | −2.27% |
| Output power (kW) | 1.105 | 1.08 | −2.26% |
| Efficiency | 91.3% | 92% | 0.76% |
| Copper (mm$^3$) | 228.6 | 228.9 | 0.13% |
| Fe (mm$^3$) | 1543 | 1249 | −19% |
| PM (mm$^3$) | 1235 | 1040 | −15.8% |

## 6. Conclusions

In this paper, modified initial design procedure and economic optimization design were studied to improve the design efficiency and economy of an AFPMSG. To improve the accuracy and efficiency of initial design, the MEC method and static magnetic field FEA were combined in the initial design procedure, which solved the problems of inaccuracy of electromagnetic parameters and too many iteration times in the MEC method, by employing the advantages of the static magnetic field FEA (i.e., accuracy and less time-consumption). For the economic optimization design, the PM material volume model, which concerned the cost of the AFPMSG the most, was derived and taken as the optimization object. The sensitivity analysis was introduced and used to distinguish and sort the performance influence degree of the main structure parameters in the model for further optimization calculation. Considering the different influence degrees and multi-parameters' interactions, the improved GA, which combined the simulated annealing method and father-offspring hybrid selection method, was studied and implemented to search for the best solution of economic optimization.

Employing the initial design procedure and economic optimization design, a 1 kW 300 r/min AFPMSG with two outer rotors and an inner coreless stator was designed and optimized on the condition of keeping the electrical performance meeting the requirements. The results show that the output voltage of the AFPMSG was very close to the sinusoidal waveform, and the torque ripple was small, which were in accordance with the structure characteristic of the AFPMSG. Moreover, the effective material cost was reduced, especially the rotor yoke and PM, while the electrical performance of the AFPMSG changed little. The 3D FEA of the generator designed by the procedure proposed in the paper, and comparison before and after optimization, illustrates the validity of the study in the paper. The improved design and analysis proposed in this paper are beneficial to the improvement and further understanding of the AFPSMG design. In the future, thermal behavior, control, other optimizations for application conditions, a specific detailed design for prototype manufacture, and a prototype experiment will be considered in the subsequent research process.

**Author Contributions:** W.W. and Y.W. designed the main parts of the research, including objective function establishment and optimal methodology design. S.Z. was responsible for guidance, a number of key suggestions, and manuscript editing. J.G. and H.M. mainly contributed to the writing of the paper and were also responsible for project administration. H.L. and G.Z. made contributions to simulations in Ansys Maxwell and gave the final approval of the version to be published.

**Funding:** This research received no external funding.

**Acknowledgments:** The authors would like to acknowledge the technological and financial support from "The National Science and Technology Support Program (supported by the Ministry of Science and Technology of P.R.C. No. 2014BAC01B05)", "Key projects of PLA (BY113C001)", "Army Funding for military graduate students (2016JY486)", and "2017 Military Logistics Open Research and Research Projects (AY117J004)". The authors are grateful that comments and suggestions provided by the anonymous reviewers and editor helped to improve the quality of the paper.

**Conflicts of Interest:** The authors declare no conflicts of interest.

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
