# Peer review of "Sensitivity Analysis and Optimal Design of a Stator Coreless Axial Flux Permanent Magnet Synchronous Generator"

_sustainability, doi:10.3390/su11051414_

Round 1

Reviewer 1 Report

Summary:

In the submitted paper, the authors have presented the design of a stator coreless axial flux permanent magnet synchronous generator (AFPMSG) considering the efficiency and economy, by the use of a field-circuit optimization method for the initial design in order to increase the design accuracy

and operability.

Recommendation: Minor revisions

Review: Overall the proposed approach is supported by the theoretical analysis and some simulation scenarios, which show the potential of the proposed approach in selected conditions. Although the proposed study is promising, there are some minor concerns regarding the details of the presented method.

-      In order to make the proposed method understandable, authors are encouraged to add a schematic overview of the whole paper.

-      The authors claim that “Analytical design and finite element method are either inaccurate or complex and time-consuming.” Such methods are widely used both in industry and academia. Please justify where do you base this strong statement.

-      Figure 8 is not readable and needs to be updated to improve its quality.

-      It would add value to present also visually the improvement compared to the initial generator, which is currently only shown at Table 3.

-      A minor issue is how the provided design can affect the actual operation of the generator, which is not straightforward from the provide results.

-      A time simulation of the AFPMSG needs to be included, in order to show that the designed system can actually operate and show its behavior against a realistic profile.

-      In order to make the conclusion section more clear, authors are highly encouraged to include the point-by-point findings of this article. The current conclusion is written very wide and it is not easy to maintain the key findings.

-      Please talk about the future work briefly in the conclusion section.

Author Response

Response to Reviewer 1 Comments

Point 1In order to make the proposed method understandable, authors are encouraged to add a schematic overview of the whole paper.

Response 1:

Thanks for your comments. We are sorry for our unclear expression of the method used in the paper. And as your kind suggestion, we intend to add a schematic overview of the whole paper and correct the last paragraph in part-Introduction. The correction parts and schematic overview of the paper are given below.

Considering the special requirements for the design and application, improved design procedure and economy optimization of a stator coreless AFPMSG are proposed in the paper. The field-circuit method combing the MEC method and static magnetic field FEA is used in the initial design procedure to improve the design efficiency and accuracy on condition of keeping the electrical performance unchanged. Active PM material volume model concerning the cost of AFPMSG most is derived for the economical optimal design. In order to find out the influence degree of each parameter, sensitivity analysis on the main design parameters to the performance and material volume consumption of AFPMSG are carried out. Considering the main design parameters non-linear interaction and different influence degree to the output performance, father off-spring selection and simulated annealing method are applied in the GA to improve the design accuracy, globality and fast convergence. Improved GA is used to perform the calculation of AFPMSG economy optimization model for the best solution to reduce active material volume cost and improve the economy. The schematic overview of this paper is showed in Fig.1.

Fig.1. The schematic overview of this paper

Thank you again for your comments which is important for me to know the shortcomings and do better in the future work!

Point 2The authors claim that “Analytical design and finite element method are either inaccurate or complex and time-consuming.” Such methods are widely used both in industry and academia. Please justify where do you base this strong statement.

Response 2

Thanks for your comments, what you have mentioned is really something that needs to be clarified to make the article more rigorous. The design and analysis of AFPMSG have been well developed, and three main methods and their combination have been widely used both in industry and academia. The three main methods are analytical method, magnetic equivalent circuit (MEC) method and finite element method.

Analytical method can solve Maxwell equations under specific assumptions and simplified conditions, and achieve magnetic field analysis and calculation with certain accuracy, saving a lot of calculation time. However, factors such as saturation of magnetic circuit, stator slot and magnetic leakage are still difficult to accurately reflect in the analytical formulas. Oversimplification may lead to low calculation accuracy. Reference [1][2][3] give the detail of analytical method application.

The special structure of axial permanent magnet motor makes the distribution of flux density along radial and axial direction reflect two independent 3D effects, "Bending Effect" and "Edge Effect". Three-dimensional finite element analysis can take into account both effects to achieve high precision magnetic field analysis, but it is difficult to avoid long calculation time, and inconvenient to apply to the initial and optimal design with a variety of parameters varying in a large range. 3D finite element method is applied in [4][5].

The MEC method used in the paper adopts the analogy method of "magnetic circuit" and "circuit". Considering the saturation of magnetic circuit, the nonlinearity of ferromagnetic material and the mutual influence of permanent magnet magnetic field and armature reactive magnetic field, the magnetic circuit equilibrium equation is established by calculating the reluctance of each magnetic circuit link, and the magnetic field distribution of the motor is obtained by solving the equation, as shown in [6][7]. The calculation time is less than that of finite element method, and the calculation accuracy is generally higher than that of analytical method.

In summary, all the methods used in the design and analysis of motor is to improve the design efficiency and accuracy. Although the main methods is widely used, each of them has its own advantages and disadvantages, and is suitable for different situation. Besides, each method also can be improved and adjusted for better design. Considering the structure and magnetic field distribution of AFPMSG proposed in the paper, the MEC method combining the static magnetic field finite element analysis is adopted to improve the design efficiency and accuracy.

To make the paper being more rigorous and understandable, sentences are corrected and added in third paragraph 3 in section - Introduction and section 2.2, as shown below.

To improve the design efficiency and accuracy, hybrid methods is used to the design, analysis and calculation of APFMSG in [18] [19]. However, the special structure and oversimplification would lower the design accuracy for analytical method, and 3D finite element method takes time to do the calculation and is inconvenient for design and analysis with multi-parameters varying in a large range. The calculation time and accuracy of MEC method are moderate and between the analytical method and finite element method. As the blossom of computer-aid technologies, new or improved design methods are still under research to make the design more realistic.

In the design procedure of AFPMSG as discussed in the previous part, some design parameters are determined by experience and estimating. Considering the characteristics of the structure and magnetic field distribution, the analytical design and finite element method may be either inaccurate for special magnetic field distribution or time-consuming for too much parameters to simulation. In this paper, the MEC method combining the static magnetic field FEA is adopted to improve the design efficiency and accuracy. The way used in the paper makes full use of the advantages of accuracy and less time-consuming of static magnetic field FEA to both solve the electromagnetic parameters’ inaccuracy problem and reduce the number of iterations, in MEC method. The MEC method in this paper converts the non-uniform spatial magnetic field into the equivalent multi-section magnetic circuit, and approximates that the magnetic flux in each section is uniformly distributed along the cross-section and length, and transforms the calculation of magnetic field into the calculation of magnetic circuit [8].

[1]     Bumby J R , Martin R , Mueller M A , et al. Electromagnetic design of axial-flux permanent magnet machines[J]. IEE Proceedings-Electric Power Applications, 2004, 151(2):151-0.

[2]     Huang Y , Ge B , Dong J , et al. 3-D Analytical Modeling of No-Load Magnetic Field of Ironless Axial Flux Permanent Magnet Machine[J]. IEEE Transactions on Magnetics, 2012, 48(11):2929-2932.

[3]     Virtič P, Vražić M, Papa G. Design of an Axial Flux Permanent Magnet Synchronous Machine Using Analytical Method and Evolutionary Optimization[J]. IEEE Transactions on Energy Conversion, 2016, 31(1):150-158.

[4]     De la Barrière, O, Hlioui S, Ben Ahmed H, et al. 3-D Formal Resolution of Maxwell Equations for the Computation of the No-Load Flux in an Axial Flux Permanent-Magnet Synchronous Machine[J]. IEEE Transactions on Magnetics, 2011, 48(1):128-136.

[5]     Chan T F, Wang W, Lai L L. Magnetic Field in a Transverse- and Axial-Flux Permanent Magnet Synchronous Generator From 3-D FEA[J]. IEEE Transactions on Magnetics, 2012, 48(2):1055-1058.

[6]     Abbaszadeh K , Maroufian S S . [IEEE 2013 21st Iranian Conference on Electrical Engineering (ICEE) - Mashhad, Iran (2013.05.14-2013.05.16)] 2013 21st Iranian Conference on Electrical Engineering (ICEE) - Axial flux permanent magnet motor modeling using magnetic equivalent circuit[J]. 2013:1-6.

[7]     Tao Z, Huang Y, Dong J, et al. Design and modeling of axial flux permanent magnet machine with yokeless and segment armature using magnetic equivalent circuit[C]// International Conference on Electrical Machines & Systems. 2014.

Point 3:  Figure 8 is not readable and needs to be updated to improve its quality.

Response 3

Thank you for your careful review and guidance. We are sorry for the unclarity in Fig. 8 and it has been replaced.

Figure 8. Flowchart of hybrid GA.

Point 4: It would add value to present also visually the improvement compared to the initial generator, which is currently only shown at Table 3.

Response 4

Thanks for your suggestion, what you have mentioned is really something that helps a lot for the presentation of the comparison. Visualization comparison of electrical performance and material consumption comparison is added before Table 3. And the contents are inserted in third paragraph of Section 5.

Finite element analysis (FEA) is proceeded on the AFPMSG to evaluate the improved design presented in the previous sections. 3-D models of AFPMSG are developed and simulated by Ansys Maxwell 16.0 software based on the improved dimensions of AFPMSG in Table 2, as shown in Fig.7. The models before and after optimization are set and simulated with the same conditions. The electrical performance is first concerned and the consumption of main material is the target of optimization. Fig.11 shows the distribution of magnetic flux density along the circumstance and comparison of Mag B curve under one pole arc. It can be seen that the distribution of Mag B is presents a flat top wave under one pole arc and periodically distributes along the circumference direction. The average value of Mag B changes little between the initial design and optimal design. As mentioned in Section 4, the decrease of rotor yoke thickness increases the value of max flux density in the rotor yoke while the air gap flux density almost keeps unchanged under the integrated influence of air gap length and PM thickness decreases.

Figure 11. The distribution of Mag B along circumference direction and comparison of Mag B curve under one pole arc.

Three phase voltage and torque of the generator are important aspects of output performance, which are discussed in the paper. The induced voltage and torque of optimal design under rated load is achieved and compared with the initial design, as shown in Fig.12. Obviously, the output phase voltage of optimal design still has a good sinusoidal curve and torque performs smoothly. The comparison of induced voltage and torque curves shows that the output performance of AFPSMG after optimization is almost the same with the initial design, which can satisfy the requirements of design purpose. Besides, the results is consistent with the comparison of Mag B.

(a) Output induced voltages waveform                    (b) Output torque waveform

Figure 12. The comparison of AFPMSG output performance.

Figure 13. The comparison of electrical performance and main material consumption

 The economy of AFPMSG is improved through the consumption of effective material especially the PM, and the comparison of electrical performance and main material consumption are shown in Fig.13. The efficiency and output power of AFPMSG also changes little while the volume of main material reduces a lot. It is observed that the whole volume of rotor yoke and PM for AFPMSG is reduced obviously after optimization, 19% and 15.8% respectively. The results show that economic optimization of AFPMSG through the main material volume cost model and improved GA method can be achieved on the condition of keeping the electrical performance unchanged. It is clear that the 3-D FEA results confirm the economy optimization of AFPMSG. The validity of both the improved initial design procedure with MEC combining the static magnetic field FEA, and the optimal design through main material consumption model and improved GA, have been confirmed by the results and analysis in the paper, which can be applied or used as a reference for better design. In addition, the research of this paper helps a lot for further understand of AFPMSG.

Point 5: A minor issue is how the provided design can affect the actual operation of the generator, which is not straightforward from the provide results.

Response 5

Thank you for your professional question. There are two main work of this paper and the first one is that the MEC bombing the static magnetic field finite element analysis is used in the initial design procedure to improve the design accuracy and efficiency. The other one is AFPMSG economy optimization design on the condition of ensuring the electrical performance meeting the design requirements through main material consumption model and improved GA. The results through the 3D model and finite element simulation by Ansys Maxwell 16.0 have confirmed the validity of the study in the paper. The research of this paper can be helpful in the design and analysis of AFPMSG. On the one hand, the research content of this paper can be used in the actual design and manufacture of motor, improving the design accuracy and efficiency of motor and reducing the cost of motor, or as the design reference for better design. On the other, the performance sensitivity analysis on the main structure parameters helps a lot to further understand the AFPMSG.

We are sorry for not expressing clearly. To make the expression being more readability, contents below are inserted in the results analysis and conclusion.

The validity of both the improved initial design procedure with MEC combining the static magnetic field FEA, and the optimal design through main material consumption model and improved GA, have been confirmed by the results and analysis in the paper, which can be applied or used as a reference for better design. In addition, the research of this paper helps a lot for further understand of AFPMSG.

The 3-D FEA of the generator designed by the procedure proposed in the paper and comparison before and after optimization illustrates the accuracy of the study in the paper. The improved design and analysis proposed in this paper are beneficial to the improvement and further understanding of AFPSMG design.

Point 6: A time simulation of the AFPMSG needs to be included, in order to show that the designed system can actually operate and show its behavior against a realistic profile.

Response 6

Thank you for your comments. The purpose of this paper aims to improve the design accuracy and efficiency, and reduce the cost through the improved design proposed. The research in the paper is carried out one step by the other. First, the structure and basic electromagnetic relationship of the AFPMSG proposed in the paper is introduced, which is the basic thing for a study. Then, the improved initial design procedure is presented by combining the static magnetic field analysis with MEC method for obtaining some electromagnetic parameters accurately. The way used in the paper makes full use of the advantages of accuracy and less time-consuming of static magnetic field FEA to both solve the electromagnetic parameters’ inaccuracy problem and reduce the number of iterations, in MEC method. The initial design in the paper is achieved based on the rated design requirements through the improved design procedure including sizing equation, MEC method and static magnetic field FEA. The performance analysis through 3D FEA confirms that the initial design is validity and can satisfy the electrical requirements.

Second, to improve the economy of AFPMSG, the effective material consumption model is derived. Sensitivity analysis is performed to find out the influence degree of each main structure parameter. Improved GA is adopted to calculate and obtain the best solution of economy optimization. Results of simulation and comparison confirm the validity. The improved initial design procedure and economy optimization are all confirmed by 3D FEA. The 3D model is established with the real generator structure and designed dimensions. The same setup and simulation conditions are applied and enough time simulations are performed to obtain the results. In conclusion, the advantages of improved design procedure and GA as well as the results show that the design proposed in the paper can be proved to be validity and is practically operable. What is said in the commentary has been analysed in the article. To make it clear, contents below are insert into the paper. We hope that the understanding of the comment is correct and our answers can satisfy you.

In the design procedure of AFPMSG as discussed in the previous part, some design parameters are determined by experience and estimating. Considering the characteristics of the structure and magnetic field distribution, the analytical design and finite element method may be either inaccurate for special magnetic field distribution or time-consuming for too much parameters to simulation. In this paper, the MEC method combining the static magnetic field FEA is adopted to improve the design efficiency and accuracy. The way used in the paper makes full use of the advantages of accuracy and less time-consuming of static magnetic field FEA to both solve the electromagnetic parameters’ inaccuracy problem and reduce the number of iterations, in MEC method. The MEC method in this paper converts the non-uniform spatial magnetic field into the equivalent multi-section magnetic circuit, and approximates that the magnetic flux in each section is uniformly distributed` along the cross-section and length, and transforms the calculation of magnetic field into the calculation of magnetic circuit [8].

Point 7: In order to make the conclusion section more clear, authors are highly encouraged to include the point-by-point findings of this article. The current conclusion is written very wide and it is not easy to maintain the key findings.

Response 7

Thank you for your comments. We are sorry for not articulating the contents clearly in conclusion. As your kind suggestion, the conclusion is revised to highlight the key findings, as shown below.

In this paper, improved initial design procedure and economy optimization design were studied to improve the design efficiency, accuracy and economy of AFPMSG. To improve the accuracy and efficiency of initial design, MEC method and static magnetic field FEA are combined in the initial design procedure, which solves the inaccuracy electromagnetic parameters and iteration times problems in the MEC method by taking advantages of the accuracy and less time-consumption of static magnetic field FEA. For the economy optimization design, the PM material volume model concerning the cost of AFPMSG most is derived and taken as the optimization object. The sensitivity analysis is introduced and used to distinguish and sort the performance influence degree of the main structure parameters in the model for further optimization calculation. Considering the different influence degree and multi-parameters’ interaction, the improved GA that combines simulated annealing method and father-son hybrid selection method is studied and implemented to search for the best solution of economy optimization .

Employing the initial design procedure and economy optimization design, a 1 kW 300r/min AFPMSG with two outer rotor and inner coreless stator is designed and optimized on condition of keeping the electrical performance meeting the requirements. The results show that the output voltage of AFPMSG is very closed to the sinusoidal waveform and the torque ripple is small, which is in accordance with the structure characteristic of AFPMSG. Moreover, the effective material cost is reduced especially the rotor yoke and PM while the electrical performance of AFPMSG changes little. The 3-D FEA of the generator designed by the procedure proposed in the paper and comparison before and after optimization illustrates the validity of the study in the paper. The improved design and analysis proposed in this paper are beneficial to the improvement and further understanding of AFPSMG design. In the future, thermal behavior, control and other optimization for application condition, specific detail design for prototype manufacture and prototype experiment will be considered in the subsequent research process.

Thank you again for your comments which is important for me to know the shortcomings and do better in the future work!

Point 8: Please talk about the future work briefly in the conclusion section.

Response 8

Thank you for your comments. In fact, the work done in this paper is part of a series of design and optimization. The AFPMSG studied in the paper is used in the direct drive wave power generation system, which is supported by national support project of PRC and key project of PLA. To make the generator more suitable for the real operation condition, thermal behavior, dynamic response and control to the unstable input will be considered in the design and optimization. The laboratory experiment as well as the actual sea experiment will be fully considered in the subsequent research process, so as to improve the research results. Thank you for your kindly advices and the future work will add in the conclusion briefly as shown below.

In the future, thermal behavior, control and other optimization for application condition, specific detail design for prototype manufacture and prototype experiment will be considered in the subsequent research process.”

Reviewer 2 Report

The paper is interesting. In principal the contribution sounds. However the paper abstract as well as the introductive part shall be revised and improved e.g. try to be more focus on the paper novelty. The simulation tool shall be carefully introduced to the readers with aim to facilitated the results reproduction.

Author Response

Dear reviewer:

Thank you for your comments concerning our manuscript entitled “Sensitivity Analysis and Optimal Design of Stator Coreless Axial Flux Permanent Magnet Synchronous Generator”, which has been submitted to the journal of Sustainability. Those comments are all valuable and very helpful for revising and improving our paper, as well as the important guiding significance to our researches. We have studied comments carefully and have made correction which we hope meet with approval. Revised portion are marked in red in the paper. The main corrections in the paper and the responds to the comments are list point by point as following:

Response to Reviewer Comments

Point 1The paper is interesting. In principal the contribution sounds. However the paper abstract as well as the introductive part shall be revised and improved e.g. try to be more focus on the paper novelty.

Response 1:

Thanks for your evaluation and kind suggestion. We are very sorry for our unclear expression and we intend to reformulate the abstract and introduction in the following paragraph.

Abstract:

“In this paper, improved initial design procedure and economy optimization design of a stator coreless axial flux permanent magnet synchronous generator (AFPMSG) is presented to improve the design accuracy, efficiency and economy. Static magnetic field finite element analysis (FEA) is combined to solve the inaccuracy elector-magnetic parameters and reduce the iteration times in magnetic equivalent circuit (MEC) method. The design accuracy and efficiency of initial design is improved by the combination of MEC method and Static magnetic field FEA. For the economy optimization, the permanent material (PM) volume model that concerns the cost of AFPMSG most is derived, and the influence degree of main structure parameters is distinguished and sorted by sensitivity analysis. Hybrid genetic algorithm that combing the simulated annealing and father-offspring selection method is studied and adopted to search for the best optimization solution from the different influence degree and nonlinear interaction parameters. A 1kW AFPMSG is designed and optimized via the proposed procedure and optimization design. Finally, 3-D finite-element models of the generator are simulated and compared to confirm the validity of the proposed improved design and the generator performance.”

Introduction:

In the topic of permanent magnet electrical machine construction, recent works have shown that the usage of neodymium-iron-boron (NdFeB) and types of permanent magnet motor have drastically increased over the last decades mainly due to cheaper cost and the development of non-pollution energy. Although there are a lot of categories of permanent magnet generator available, axial flux permanent magnet synchronous generator (AFPMSG) with different electromagnetic path from traditional motor is studied here. AFPMSG has the merits of both permanent magnet motor and disc type motor, which are small size, simple and compact structure, high power density and operation efficiency, easy processing and manufacturing. Great application prospect and development value have been exploiting in fields of Electric cars, large industrial equipment, wind power generation, et al. [1] [2] [3] [4].

AFPMSG can be single sided or double sided, core or core less, surface mounted or interior PM and single or multi staged configuration [5]. Among those types of AFPMSG, double sided AFPMSG with stator coreless has been widely researched and used in the industry. Compared with stator cored generator, AFPMSG without stator core have the advantages of light weight, no cogging torque caused by stator core tooth, high efficiency and simple construction. The double sided structure of APFMSG has higher torque-to-volume ratio than that of other structure [6] and the more number of pole pairs is designed, the higher ratio it will be. Besides, the negligible axial attraction force between the stator and rotor improves the reliable design of large diameter generators. However, because of the non-ferromagnetic stator core, the effective air gap length is larger and requires more active PM material to keep the magnetic field in the air gap, which increases the cost of AFPMSG. Economy optimization design for the stator coreless AFPMSG becomes an important research topic.

With the further research and extensive application of AFPMSGs, the analysis of their design and optimization has been a hotpot research field recently. The non-uniform distribution of magnetic field, large air gap and the interaction of structural parameters make the state coreless AFPMSG a multi-variable non-linear system, which brings great difficulties to its optimization design [7]. At present, the design and optimization of AFPMSG have been extensively studied and explored. The basic theory of AFPMSG including the design procedure, performance calculation and application is presented in [5] [7]. The design methods and development of performance analysis tools for axial flux permanent magnet machines is presented in [10]. Kahourzade S, S Huang et al. [11] [12] investigated the design and analysis of AFPMSG with the sizing equation. In [13] [17], analytical methods are used for the analysis and calculation of magnetic field in the generator. Study on the flux distribution of each part of the AFPMSG via magnetic equivalent circuit (MEC) is presented in [15] [16]. A new three-dimensional finite element method based on improved Maxwell equations is proposed to calculate the no-load flux of an axial permanent magnet motor in [17]. To improve the design efficiency and accuracy, hybrid methods is used to the design, analysis and calculation of APFMSG in [18] [19]. However, the special structure and oversimplification may lower the design accuracy for analytical method, and 3D finite element method takes time to do the calculation and is inconvenient for design and analysis with multi-parameters varying in a large range. The calculation time and accuracy of MEC method are moderate and between the analytical method and finite element method. And as the blossom of computer-aid technologies, new or improved design methods are still under research to make the design more realistic.

Due to complex electromagnetic processes and practical application requirements, optimal procedure is necessary to obtain a better design results or improve the performance of the motor for the special need. Recently, heuristic algorithms especially the genetic algorithm (GA) have been used in the motor optimization [20] [21]. The development of field computation and multi-objective optimal design in electromagnetics were studied in [22] and [23]. Performance optimizations including the maximum power density, low cogging torque, minimum material cost, et al. via the GA are presented in [24][25][26]. However, in these studies, the sensitivity analysis of the generator design parameters to the output performance and economy optimization is not paid enough attention, which affects the efficiency and results of motor optimization design.  

Considering the special requirements for the design and application, improved design procedure and economy optimization of a stator coreless AFPMSG are proposed in the paper. The field-circuit method combing the MEC method and static magnetic field FEA is used in the initial design procedure to improve the design efficiency and accuracy on condition of keeping the electrical performance unchanged. Active PM material volume model concerning the cost of AFPMSG most is derived for the economical optimal design. In order to find out the influence degree of each parameter, sensitivity analysis on the main design parameters to the performance and material volume consumption of AFPMSG are carried out. Considering the main design parameters non-linear interaction and different influence degree to the output performance, father off-spring selection and simulated annealing method are applied in the GA to improve the design accuracy, globality and fast convergence. Improved GA is used to perform the calculation of AFPMSG economy optimization model for the best solution to reduce active material volume cost and improve the economy. The schematic overview of this paper is showed in Fig.1.

Figure 1. The schematic overview of this paper.

In the following, basic structure and equations are introduced and improved initial design procedure of AFPMSG with the field- circuit method is given in Section. The economic optimization model that is the active PM material volume is deprived in Section , as well as the sensitivity analysis about the main design parameters to the performance and material volume consumption of AFPMSG. Section introduces the improved GA for the optimal design of APFMSG, which combines the simulated annealing and father-offspring selection method. In Section , the validity of improved initial design procedure and economy optimization design is evaluated by comparison and 3-D FEA. Finally, conclusions are drawn in Section .”

Thank you again for your comments which is important for me to know the shortcomings and do better in the future work!

Point 2: The simulation tool shall be carefully introduced to the readers with aim to facilitated the results reproduction.

Response 2:

Thank you for your kind suggestion. As you suggested, the introduction of simulation tool is added before the performance analysis in Section 2.3, as shown below.

3D finite element analysis is adopted to simulate the output performance of AFPSMG through Ansys Maxwell 16.0 software. Ansys Maxwell is the most practical electromagnetic analysis software at present, which can analyze and calculate various electromagnetic fields and multi-state system problems with its advanced adaptive meshing technology and convenient self-defined material library[31]. The main steps of simulation analysis are shown in Fig.6[32]. In this paper, the establishment of model is based on the design value of generator structure dimension. The other settings and conditions are set as the normal or rated value.

Figure 6. The Simulation and design flow chart of Ansys Maxwell

The performance analysis of the generator mainly focuses on the analysis and research of the induced electromotive voltage, torque and efficiency of the generator under load to verify the design results. The initial AFPMSG simulation model is established based on the initial design structure dimension values, as shown in Fig. 7(a). The initial operation condition is under rated constant speed 300r/min with three-phase symmetrical rated resistive load. The output performance curve including the induced voltage and torque of AFPMSG can be obtained through simulation as shown in Fig. 8.”

[1].    Ansoft Company. Ansoft User Manual [M]. Ansoft Company 2007.

[2].    Zhao Bo, Zhang Hongliang. Application of Ansoft 12 in Engineering Electromagnetic Field [M]. China Water Resources and Hydropower Press, 2010.

Thank you again for your kind suggestion. We will improve our wring in the future work!

We tried our best to improve the manuscript and made some changes in the manuscript according to the comments. What’s more, after careful examination and modification, we also made some small changes on grammatical errors and so on. We appreciate for editors/reviewers’ warm work earnestly, and hope that the correction will meet with approval.

Once again, thank you very much for your comments and suggestions.

Thank you and best regards.

                                            Yours sincerely

                                             Shaoqi Zhou

Reviewer 3 Report

The novelties introduced in the paper are not highlighted.

The paper proposes the design of a coreless axial flux generator.

The type of electrical machine is well known in literature and also the proposal design procedure and the optimization can be considered standard.

1) The authors are invited to emphasize all the novelties of the paper. 

2) How the thermal behavior of the machine is verified?

3) How the magnet demagnetization is checked? 

Author Response

Dear reviewer:

Thank you for your comments concerning our manuscript entitled “Sensitivity Analysis and Optimal Design of Stator Coreless Axial Flux Permanent Magnet Synchronous Generator”, which has been submitted to the journal of Sustainability. Those comments are all valuable and very helpful for revising and improving our paper, as well as the important guiding significance to our researches. We have studied comments carefully and have made correction which we hope meet with approval. Revised portion are marked in red in the paper. The main corrections in the paper and the responds to the comments are list point by point as following:

Response to Reviewer Comments

Point 1: The authors are invited to emphasize all the novelties of the paper.

Response 1:

Thank you for your comments. We are sorry for not articulating the novelties of the paper. The purpose of this paper aims to improve the design accuracy and efficiency, and reduce the cost through the improved design proposed. The novelties work in the paper can be divided into two aspect. The first one is that static magnetic field finite element analysis (FEA) is applied in the magnetic equivalent circuit (MEC) method to solve the inaccuracy electromagnetic parameters and reduce the iteration times. The field-circuit method combines the MEC method and static magnetic field FEA is used in the initial design procedure to improve the design accuracy and efficiency. The other one is that the performance influence degree of the main structure parameters is considered in the optimization of active PM material volume cost, and improved GA that combines the simulated annealing method and father-son hybrid selection method is used to search the best solution from the different influence degree and interaction parameters. Thank you for your advice, and we revised the abstract and conclusion to make the expression of our work clearly, as shown below.

Abstract:

In this paper, improved initial design procedure and economy optimization design of a stator coreless axial flux permanent magnet synchronous generator (AFPMSG) is presented to improve the design accuracy, efficiency and economy. Static magnetic field finite element analysis (FEA) is combined to solve the inaccuracy electromagnetic parameters and reduce the iteration times in magnetic equivalent circuit (MEC) method. The accuracy and efficiency of initial design is improved by the combination of MEC method and Static magnetic field FEA. For the economy optimization, the permanent material (PM) volume model that concerns the cost of AFPMSG most is derived, and the performance influence degree of main structure parameters is distinguished and sorted by sensitivity analysis. Hybrid genetic algorithm that combines the simulated annealing and father-offspring selection method is studied and adopted to search for the best optimization solution from the different influence degree and nonlinear interaction parameters. A 1kW AFPMSG is designed and optimized via the proposed design procedure and optimization design. Finally, 3-D finite-element models of the generator are simulated and compared to confirm the validity of the proposed improved design and the generator performance.

Conclusion:

In this paper, improved initial design procedure and economy optimization design were studied to improve the design efficiency and economy of AFPMSG. To improve the accuracy and efficiency of initial design, MEC method and static magnetic field FEA are combined in the initial design procedure, which solves the inaccuracy electromagnetic parameters and iteration times problems in the MEC method by taking advantages of the accuracy and less time-consumption of static magnetic field FEA. For the economy optimization design, the PM material volume model concerning the cost of AFPMSG most is derived and taken as the optimization object. The sensitivity analysis is introduced and used to distinguish and sort the performance influence degree of the main structure parameters in the model for further optimization calculation. Considering the different influence degree and multi-parameters’ interaction, the improved GA that combines simulated annealing method and father-son hybrid selection method is studied and implemented to search for the best solution of economy optimization .

Employing the initial design procedure and economy optimization design, a 1 kW 300r/min AFPMSG with two outer rotor and inner coreless stator is designed and optimized on condition of keeping the electrical performance meeting the requirements. The results show that the output voltage of AFPMSG is very closed to the sinusoidal waveform and the torque ripple is small, which is in accordance with the structure characteristic of AFPMSG. Moreover, the effective material cost is reduced especially the rotor yoke and PM while the electrical performance of AFPMSG changes little. The 3-D FEA of the generator designed by the procedure proposed in the paper and comparison before and after optimization illustrates the validity of the study in the paper. The improved design and analysis proposed in this paper are beneficial to the improvement and further understanding of AFPSMG design. In the future, thermal behavior, control and other optimization for application condition, specific detail design for prototype manufacture and prototype experiment will be considered in the subsequent research process.

Thank you again for your comments which is important for me to know the shortcomings and do better in the future work!

Point 2: How the thermal behavior of the machine is verified?

Response 2:

Thank you for professional question. The purpose of this paper is to improve the design accuracy, efficiency and economy of AFPMSG. All the work is focused on the goal. The main works of this paper are to find ways to improve the accuracy and efficiency in initial design procedure, and to proper optimize the economy of AFPMSG with electrical performance unchanged. The results confirm the validity of the research in the paper and the research can be use in the real design or as a reference to the design and analysis of AFPSMG. The generator proposed in the paper has two rotors with PM fixed on it and they can be taken as fans when the rotors rotate, which makes the motor have better heat dissipation effect. In my opinion, the thermal influence is little in normal conditions and thermal behavior has little to do with the research in the paper. However, to make the research be more perfect, further researches including the thermal behavior should be done. The thermal behavior of the machine is another important factor that affect the generator performance. Thermal research of machine involves various contents, which can be a big research field. Thank you for your kindly reminding, and thermal behavior will be fully considered in the subsequent research process.

Thank you for offering another research idea. We hope that the understanding of the comment is correct and our answers can satisfy you.

Point 3: How the magnet demagnetization is checked?

Response 3:

Thank you for your comments. The magnetic source of the stator coreless AFPMSG proposed in the paper is the permanent magnet. And due to the special structure that stator is made by the windings and non-ferromagnetic materials, the air gap is larger than other type generators, which requires more PM to establish the magnetic field. On the same way, the armature reaction has little effect on the magnetic field, let alone the demagnetization of permanent magnets. That is to say, the magnetic field of generator proposed in the paper is relative stable and the magnet will not be affected by the armature reaction expect for the extreme conditions. The factors that cause the magnet demagnetization are more likely to be the environment factors such as the thermal influence, saline alkaline corrosion and so on. For special application, the influence data of environmental factors on permanent magnets can be obtained from the manufacturer or through experiments. The influence should be fully considered during the design stage. The influence data should be used to correct the design value of magnetic and import into the simulation software for correcting during the simulation.

The study of magnet demagnetization is another important part of permanent magnet machine research for practical application. And the magnet demagnetization deserve well studied for the application. As far as I can see, the magnet demagnetization should be considered according to the specific conditions. The study in this paper is not for a special situation and the factors that cause the demagnetization is not certain. Thus, the magnet demagnetization is not considered in the paper. When the generator is for a certain application or magnetic demagnetization of generator is a thematic research, contents including the influence factors, the influence mechanism as well as how to check the magnet demagnetization, protect and so on will be carefully studied.

Thank you again for sharing the ideas to improve the research. We hope that our understanding of your comments is correct and our answer makes sense to you.

We tried our best to improve the manuscript and made some changes in the manuscript according to the comments. What’s more, after careful examination and modification, we also made some small changes on grammatical errors and so on. We appreciate for editors/reviewers’ warm work earnestly, and hope that the correction will meet with approval.

Once again, thank you very much for your comments and suggestions.

Thank you and best regards.

                                            Yours sincerely

                                             Shaoqi Zhou

Round 2

Reviewer 3 Report

- It is preferrable plot the torque (Fig.8b) vs the rotor angle and not in the time domain;

- One of the problem of coreless (abd in particular for the electricl machine with low iron part) is inherent the low value of synchronous inductance. How the authors solve thisproblem?

Author Response

Dear reviewer:

Thank you for your comments, and they are all valuable and very helpful for revising and improving our paper. Your comments are carefully studied and corrections have been made, which we hope our understanding of your comment is correct and the answer will meet with approval. Revised portion are marked in yellow in the paper. The main corrections in the paper and the responds to the comments are list point by point as following:

Response to Reviewer Comments

Point 1: It is preferrable plot the torque (Fig.8b) vs the rotor angle and not in the time domain.

Response 1:

Thank you for your careful review and guidance. Fig. 8 here shows the result of initial design and is to verify the electrical performance. The relationship between torque and rotor angle can illustrates the output results better. As your suggestion, we replace the Fig.8 (b) with picture shown below, and the contents for explanation inserted in the paper.

(b) Torque against the rotor angle

 “It is evident that the AFPMSG has a sinusoidal induced voltage and relative stable electromagnetic torque output. The output induced voltage is stable with low harmonics. The electromagnetic torque ripple caused by eddy current and error is small, and is overall balance in a cycle, which can be neglected. The maximum amplitude of fluctuation is about 1.5Nm.

Thank you again for your kind suggestion. We will improve our wring in the future work!

Point 2: One of the problem of coreless (abd in particular for the electricl machine with low iron part) is inherent the low value of synchronous inductance. How the authors solve this problem?

Response 2:

Thanks for your valuable question and what you have mentioned is really something that need to be discussed. The synchronous inductance of stator coreless AFPMSG is lower than other topologies’ due to the non-ferromagnetic materials that makes the stator. The synchronous inductance has great influence on the generator performance. One of the influence is that the output voltage varies almost linearly with the load current, which means that the output of AFPMSG changes with the input condition and it is hard to make the change. The traditional method of adjusting the air gap magnetic field by adjusting the direct axis demagnetizing current has limited magnetic weakening ability and cannot maintain the stable output of voltage. It need much bigger demagnetizing current to realize the adjusting purpose, which adds the loss, reduce the efficient and brings the risk of irreversible demagnetization of permanent magnets. As a motor, the control problem of low synchronous inductance machine is more difficult. There are two ways to solve the problem. The first one is working on the generator itself to change the situation, such as optimal design to improve the synchronous inductance, adjustment by hybrid excitation method and so on. The second way is developing corresponding controllers and power converters. The study of synchronous inductance influence and its solution is a popular topic in the optimization and application of axial flux permanent magnet synchronous machines.

In this paper, the initial design procedure is optimized by the combination of MEC method and static field FEA, and the economy of AFPMSG is improved by the PM volume model and improved GA. The parameter of AFPMSG is design by design procedure based on the design requirements on the purpose of electrical performance.  The optimization calculation also based on the condition of maintaining the electrical performance. The synchronous inductance has been considered in the design process. But little attention has been paid to the impact of low synchronous inductance and its solutions. It will be fully considered in my future work and helpful to improve my study.

Thank you for your kindly comment and we hope that the understanding of the comment is correct and our answers can satisfy you.

We tried our best to improve the manuscript and made some changes in the manuscript according to the comments. We appreciate for reviewer’s warm work earnestly, and hope that the correction will meet with approval.

Once again, thank you very much for your comments and suggestions.

Thank you and best regards.

                                            Yours sincerely

                                             Shaoqi Zhou
